# COMPLEXITY-DRIVEN POLICY OPTIMIZATION

## ABSTRACT

Policy gradient methods often balance exploitation and exploration via entropy maximization. However, maximizing entropy pushes the policy towards a uniform random distribution, which represents an unstructured and sometimes inefficient exploration strategy. In this work, we propose replacing the entropy bonus with a more robust complexity bonus. In particular, we adopt a measure of complexity, defined as the product of Shannon entropy and disequilibrium, where the latter quantifies the distance from the uniform distribution. This regularizer encourages policies that balance stochasticity (high entropy) with structure (high disequilibrium), guiding agents toward regimes where useful, non-trivial behaviors can emerge. Such behaviors arise because the regularizer suppresses both extremes, e.g., maximal disorder and complete order, creating pressure for agents to discover structured yet adaptable strategies. Starting from Proximal Policy Optimization (PPO), we introduce Complexity-Driven Policy Optimization (CDPO), a new learning algorithm that replaces entropy with complexity. We show empirically across a range of discrete action space tasks that CDPO is more robust to the choice of the complexity coefficient than PPO is with the entropy coefficient, especially in environments requiring greater exploration.

## 1 INTRODUCTION

Reinforcement Learning (RL) has achieved significant results in numerous sequential decision-making problems, including board games (Silver et al., 2017) and video games (Mnih et al., 2015), robotic control (Tang et al., 2025), protein design (Angermueller et al., 2020), and human alignment in language modeling (Ouyang et al., 2022). However, for certain real-world tasks, the state space might be incredibly vast, and the reward function too sparse. Balancing exploration and exploitation and managing such *complexity* is one of the key open challenges in the field (Sutton & Barto, 2018).

Ironically, defining complexity is very complex (Mitchell, 2009), which may partly explain why it has received less attention than properties such as quality and diversity, despite its essential role in learning and generalization (Havrilla et al., 2024). In the context of RL, the exploration-exploitation trade-off plays a fundamental role in pursuing robustness and generalization (Jiang et al., 2023). To counter-balance the convergent effect of reward maximization, research has mainly focused on inducing exploration through additional intrinsic rewards (Ladosz et al., 2022) or, more naively, by maximizing the policy entropy. Indeed, several policy gradient methods, such as Proximal Policy Optimization (PPO) (Schulman et al., 2017b), often include entropy regularization. This can help with exploration by encouraging stochastic policies and preventing deterministic behavior, and in certain scenarios, it can also facilitate the optimization process (Ahmed et al., 2019). However, especially for PPO, its real utility is disputed, and finding the correct scaling factor for the entropy loss is not trivial (Andrychowicz et al., 2021); in addition, maximizing entropy pushes the policy towards a uniform random distribution regardless of the current nature of the policy and the actual need for exploration, which might result in a less efficient learning strategy (Zhang et al., 2025a).

In this work, we propose a new regularization term based on complexity, defined as the product of Shannon entropy and disequilibrium, rather than the sole entropy. This measure, usually referred to as López-Ruiz, Mancini, and Calbet (LMC) complexity measure (López-Ruiz et al., 1995), was developed to evaluate the complexity of physical systems at a given scale and is characterized by an interplay between the information stored by the system (its entropy) and the distance from equipartition (its disequilibrium). By acting as a regularizer, this complexity suppresses both extremes (e.g., total disorder and perfect order), thereby exerting pressure on agents to identify strategies that bal-

ance structure with adaptability. We then define Complexity-Driven Policy Optimization (CDPO), a new learning algorithm based on PPO that replaces its entropy bonus with a complexity term. By scaling the policy entropy with its disequilibrium, CDPO is capable of promoting divergence when the policy becomes too deterministic, while also promoting convergence when the policy is too random. The proposed mechanism can be applied to other policy optimization methods that rely on entropy-based exploration. Experiments on several classic RL environments, as well as a novel version of CartPole with fine-grained complexity control, demonstrate that CDPO is more robust to the choice of loss scaling factor than entropy-regularized PPO and achieves competitive results across settings that require different levels of exploration.

## 2 RELATED WORK

### 2.1 ENTROPY REGULARIZATION IN REINFORCEMENT LEARNING

The concept of entropy has been extensively used in RL to incentivize exploration. In particular, there exist two main strategies to optimize for entropy: the so-called maximum entropy framework, where the policy entropy and the return are summed and jointly maximized; and the entropy regularization through a separate cost function. The former has been studied in various RL domains, from inverse reinforcement learning (Ziebart et al., 2008) to stochastic optimal control (Toussaint, 2009; Rawlik et al., 2013) and especially off-policy algorithms (Haarnoja et al., 2017; 2018). The latter has been adopted in the context of an on-policy setting (Williams & Peng, 1991; Mnih et al., 2016; Schulman et al., 2017b) and a combination of value-based and policy-based RL (O'Donoghue et al., 2017), and Schulman et al. (2017a) proved them to be equivalent under entropy regularization. While entropy regularization can make the policy optimization landscape smoother (Ahmed et al., 2019), entropy-regularized methods may fail to converge to a fixed point (Neu et al., 2017). In addition, entropy is not always helpful (Liu et al., 2021; Zhang et al., 2025a), and its coefficient plays a relevant role in its effectiveness (Andrychowicz et al., 2021). Because of this, several modifications have been developed in the past. For example, Zhao et al. (2019) propose a reward-weighted entropy objective in the maximum entropy framework to solve multi-goal RL. In an off-policy setting, Han & Sung (2021b) suggest training the Q-function to minimize entropy (visiting states with low entropy), while maintaining the policy entropy maximization term in the policy update. It is also possible to maximize the entropy of the weighted sum of the current policy action distribution and the sample action distribution from the replay buffer (Han & Sung, 2021a), or to maximize the entropy of the state distribution induced by the current policy (Hazan et al., 2019). In this work, we preserve the elegance and simplicity of (policy-based) entropy regularization, addressing its limitations by scaling it with the policy disequilibrium.

### 2.2 COMPLEXITY AND REINFORCEMENT LEARNING

Recent research explores the idea that exposing models to structured complexity can lead to the development of more generalized and intelligent behaviors (Havrilla et al., 2024). Zhang et al. (2025b) investigate this phenomenon by pretraining large language models on synthetic data generated by elementary cellular automata, finding that models trained on data from systems at the "edge of chaos" (Langton, 1990), poised between order and disorder, exhibit superior performance on downstream reasoning tasks. Similarly, in the context of supervised learning, Zhang et al. (2021) demonstrate that a network's generalization capability is maximized when its internal weight dynamics are poised at the critical boundary separating ordered and chaotic regimes. Their work shows how standard training algorithms naturally push the network towards this edge and how regularization can be precisely tuned to maintain the network in this optimal state. Complexity plays a key role in reinforcement learning as well. To enhance the robustness and stability of the learned policy, Young & Pugeault (2024) use the Maximal Lyapunov Exponent (MLE) (Lyapunov, 1992) from chaos theory to demonstrate that policies can be highly sensitive to initial conditions, where small perturbations in the state lead to vastly different long-term trajectories. To mitigate this instability, they propose a novel regularization term that directly penalizes the MLE, encouraging the agent to learn more stable and predictable dynamics. Conversely, our work builds on the idea that complexity-based regularization should not aim for a predictable dynamics, but for a more stable and less random exploration.

## 3 BACKGROUND

### 3.1 REINFORCEMENT LEARNING AND PROXIMAL POLICY OPTIMIZATION

We focus on Markov Decision Processes (MDPs) expressed by the tuple $(\mathcal{S}, \mathcal{A}, P, r, \gamma)$, where $\mathcal{S}$ is the state space, $\mathcal{A}$ is the discrete action space, $P(s'|s,a)$ is the transition probability between states, $r(s,a)$ is the reward associated to that transition, and $\gamma$ is the discount factor. The RL agent uses a $\boldsymbol{\theta}$-parameterized policy $\pi_{\boldsymbol{\theta}}(a|s)$, i.e., a mapping from the state space to distributions over actions, to maximize the discounted return $\mathbb{E}_{\pi_{\boldsymbol{\theta}}}[\sum_{t=0}^{T} \gamma^t r(s_t, a_t)]$. To learn $\pi_{\boldsymbol{\theta}}$, many policy gradient methods work by applying a stochastic gradient ascent algorithm over the estimator of the gradient provided by the policy gradient theorem (Sutton et al., 1999): $\nabla_{\boldsymbol{\theta}} J(\boldsymbol{\theta}) \propto \mathbb{E}_{\pi_{\boldsymbol{\theta}}}[\nabla_{\boldsymbol{\theta}} \log \pi_{\boldsymbol{\theta}}(a_t|s_t) \hat{A}_t]$, where $\hat{A}_t$ is an estimator of the advantage function at timestep $t$, which quantifies, given $s_t$, how much better $a_t$ is with respect to other actions on average. Several policy gradient algorithms have been derived from the policy gradient theorem. However, directly updating the policy by a step of gradient ascent through a sample-based estimate can suffer from large variance and instability, and methods that constrain the update size have usually been preferred (Schulman et al., 2015). Proximal Policy Optimization (PPO) does so by maximizing a clipped surrogate objective that provides a stable and reliable update and allows for multiple iterations over the same samples (Schulman et al., 2017b):

$$L_t^{CLIP}(\boldsymbol{\theta}) = \hat{\mathbb{E}}_t\left[\min\left(\frac{\pi_{\boldsymbol{\theta}}(a_t|s_t)}{\pi_{\boldsymbol{\theta}_{old}}(a_t|s_t)}\hat{A}_t, \text{clip}\left(\frac{\pi_{\boldsymbol{\theta}}(a_t|s_t)}{\pi_{\boldsymbol{\theta}_{old}}(a_t|s_t)}, 1-\epsilon, 1+\epsilon\right)\hat{A}_t\right)\right], \quad (1)$$

where $\pi_{\boldsymbol{\theta}_{old}}$ is the policy before the update from which the action has been sampled, and $\epsilon$ is the clipping factor. The advantage $\hat{A}_t$ is computed as the generalized advantage estimation (Schulman et al., 2016) and makes use of a state-value function approximator $V_{\boldsymbol{\theta}}$ that learns to minimize $L_t^{VF}(\boldsymbol{\theta}) = \hat{\mathbb{E}}_t[(V_{\boldsymbol{\theta}}(s_t) - V_t^{targ})^2]$. Finally, PPO also includes an entropy bonus $S[\pi_{\boldsymbol{\theta}}]$ to ensure sufficient exploration; thus, the overall objective becomes:

$$L_t(\boldsymbol{\theta}) = \mathbb{E}_t\left[L_t^{CLIP}(\boldsymbol{\theta}) - c_{vf}L_t^{VF}(\boldsymbol{\theta}) + c_{reg}S[\pi_{\boldsymbol{\theta}}](s_t)\right], \quad (2)$$

with $c_{vf}, c_{reg}$ coefficients for the value-function loss and the entropy term, respectively.

### 3.2 LMC COMPLEXITY

López-Ruiz, Mancini, and Calbert (LMC) complexity is a measure based on a probabilistic description of physical systems (López-Ruiz et al., 1995). Different from other complexity measures that analyze streams of data (e.g., Grassberger (1986); Lempel & Ziv (1976)), it is based on the statistical description of systems at a given scale, and on the idea that a system is said to be *complex* when it does not conform to patterns regarded as *simple*. In particular, two physical systems have simple models and, thus, ideal zero complexity: the perfect crystal, which is completely ordered and can be described with minimal information; and the isolated ideal gas, which is completely disordered and has maximal information. In contrast, all other particle systems exhibit non-zero complexity. Similarly, in the case of RL, two policies correspond to simple distributions with ideal zero complexity: a fully random policy and a deterministic policy. All other policies exhibit non-zero complexity, with increasing complexity when all actions are feasible but certain actions have a greater probability, that is, when the policy maintains a structured yet stochastic nature. Assuming a system with $N$ accessible states $\{x_1, x_2 \ldots x_N\}$, each with probability $p_i$ with $\sum_{i=1}^{N} p_i = 1$, the LMC complexity is defined as follows:

$$C = H \cdot D = \underbrace{\left(-\sum_{i=1}^{N} p_i \log p_i\right)}_{\text{Entropy}} \cdot \underbrace{\left(\sum_{i=1}^{N}\left(p_i - \frac{1}{N}\right)^2\right)}_{\text{Disequilibrium}}. \quad (3)$$

In essence, LMC complexity is characterized by an interplay between the information stored by the system and the distance from equipartition. Thus, it is equal to zero if the entropy is zero (as for the

perfect crystal or, in the case of RL, when an action has a probability of $1$.) or if the disequilibrium is zero (as for the isolated ideal gas or, in the case of RL, when all actions are equiprobable); and it is high when the system is both stochastic and structured.

# 4 METHOD

## 4.1 OVERVIEW

Incorporating entropy maximization into policy gradient methods, such as PPO, has been shown to enhance exploration and prevent the agent from converging to deterministic strategies. However, maximizing entropy unconditionally drives the policy towards a uniform distribution. If the entropy scaling factor is too high, the entropy term can dominate the policy loss, preventing the agent from finding an optimal solution. Furthermore, in environments that do not require extensive exploration, excessive entropy can unnecessarily slow convergence (Zhang et al., 2025a).

To avoid aiming at randomness too much, but still preventing determinism, we propose to replace the entropy loss with an LMC complexity loss. As introduced in Section 3.2, LMC complexity aims to avoid *simplicity*; the system must have many possible states (high entropy), and, at the same time, it must not be completely random (high disequilibrium). While entropy is maximum if all events are equiprobable and minimum if only one of them is possible, complexity is minimum if either all events are equiprobable or only one of them is possible, and has multiple maxima in between. Figure 1 illustrates its behavior for a system with 2 possible events: whenever the system is too deterministic, maximizing complexity pushes it towards randomness; and whenever the system is too random, maximizing complexity pushes it towards determinism.

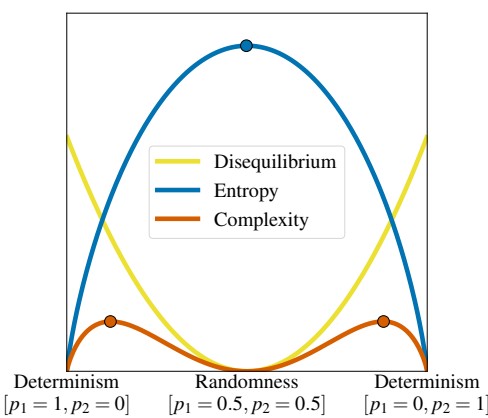

Figure 1: Disequilibrium, entropy, and complexity in a two-dimensional space. The bullet points indicate the maxima we aim to reach when optimizing for entropy or complexity.

More formally, given a discrete action space $\mathcal{A}$ and a policy $\pi_{\boldsymbol{\theta}}(a|s)$, the complexity of the policy can be defined as follows:

$$C[\pi_{\boldsymbol{\theta}}](s) = S[\pi_{\boldsymbol{\theta}}](s) \cdot D[\pi_{\boldsymbol{\theta}}](s) = \underbrace{-\sum_{a \in \mathcal{A}} \pi_{\boldsymbol{\theta}}(a|s) \log \pi_{\boldsymbol{\theta}}(a|s)}_{\text{Entropy}} \cdot \underbrace{\sum_{a \in \mathcal{A}} \left( \pi_{\boldsymbol{\theta}}(a|s) - \frac{1}{|\mathcal{A}|} \right)^2}_{\text{Disequilibrium}} \quad (4)$$

This quantity can then be optimized for by any policy gradient method in order to foster complexity rather than the sole entropy. In particular, we define here a new learning algorithm, namely Complexity-Driven Policy Optimization (CDPO), which leverages the PPO learning scheme while scaling the entropy with the policy disequilibrium, i.e., replacing its entropy term $S[\pi_{\boldsymbol{\theta}}]$ with Equation 4. In this way, CDPO moves from trying to randomize the policy to trying to maintain exploration without falling into the randomness pit. Overall, the objective function of CDPO is as follows:

$$L_t(\boldsymbol{\theta}) = \mathbb{E}_t \left[ L_t^{CLIP}(\boldsymbol{\theta}) - c_{vf} L_t^{VF}(\boldsymbol{\theta}) + c_{reg} C[\pi_{\boldsymbol{\theta}}](s_t) \right]. \quad (5)$$

From a practical standpoint, we can extend current implementations of PPO to CDPO and promote complexity rather than randomness by just computing the value of the disequilibrium and multiplying it by the entropy. In a similar way, this complexity-based mechanism can be applied to other policy approximation algorithms that rely on entropy-based exploration.

---

**Complexity-Driven Policy Optimization**

---

Input: Initialized network parameters $\boldsymbol{\theta}$, step size $\mu$

Loop for iteration $= 1, 2, \ldots$:

    Collect a set of trajectories $\mathcal{D} = \{s_i, a_i, \hat{A}_i, V_i^{targ}, \pi_{\boldsymbol{\theta}_{old}}(a_i|s_i)\}_{i=1}^{D}$ following $\pi_{\boldsymbol{\theta}}(\cdot|\cdot)$

    Loop for each policy epoch $= 1 \ldots N$:

        Split the training batch $\mathcal{D}$ into $K = \frac{D}{B}$ random minibatches

        Loop for each minibatch $k = 1 \ldots K$:

$$\hat{A}_t = \frac{\hat{A}_t - \text{mean}(\{\hat{A}\}_{i=0}^{B})}{\text{std}(\{\hat{A}\}_{i=0}^{B})} \text{ for } t = 1 \ldots B$$

$$C[\pi_{\boldsymbol{\theta}}](s_t) = \mathbb{E}_t\Big[-\sum_{a\in\mathcal{A}} \pi_{\boldsymbol{\theta}}(a|s_t)\log\pi_{\boldsymbol{\theta}}(a|s_t)\cdot\sum_{a\in\mathcal{A}}(\pi_{\boldsymbol{\theta}}(a|s_t)-\tfrac{1}{|\mathcal{A}|})^2\Big]$$

$$L_t^{CLIP}(\boldsymbol{\theta}) = \hat{\mathbb{E}}_t\Big[\min\Big(\tfrac{\pi_{\boldsymbol{\theta}}(a_t|s_t)}{\pi_{\boldsymbol{\theta}_{old}}(a_t|s_t)}\hat{A}_t, \text{clip}\Big(\tfrac{\pi_{\boldsymbol{\theta}}(a_t|s_t)}{\pi_{\boldsymbol{\theta}_{old}}(a_t|s_t)}, 1-\epsilon, 1+\epsilon\Big)\hat{A}_t\Big)\Big]$$

$$L_t^{VF}(\boldsymbol{\theta}) = \hat{\mathbb{E}}_t\big[(V_{\boldsymbol{\theta}}(s_t)-V_t^{targ})^2\big]$$

$$L_t(\boldsymbol{\theta}) = \mathbb{E}_t[L_t^{CLIP}(\boldsymbol{\theta}) - c_{vf}L_t^{VF}(\boldsymbol{\theta}) + c_{reg}C[\pi_{\boldsymbol{\theta}}](s_t)]$$

$$\theta = \theta + \mu\nabla_{\boldsymbol{\theta}}L_t(\boldsymbol{\theta})$$

---

### 4.2 GRADIENT ANALYSIS

To explain why the complexity bonus is more robust than the entropy bonus, we now examine the respective gradients. We will show that, unlike the entropy landscape, the complexity optimization landscape contains multiple maxima and lowers the probability of deterministic and purely random behaviors.

The gradient of the entropy bonus is $\nabla_{\boldsymbol{\theta}}S[\pi_{\boldsymbol{\theta}}](s) = \sum_a -(\log\pi_{\boldsymbol{\theta}}(a|s)+1)\nabla_{\boldsymbol{\theta}}\pi_{\boldsymbol{\theta}}(a|s)$. When subject to the probability constraint $\sum_a \pi_{\boldsymbol{\theta}}(a|s) = 1$, its maximum lies in $\pi_{\boldsymbol{\theta}}(a|s) = \frac{1}{|\mathcal{A}|}, \forall a \in \mathcal{A}$ (the proof is reported in Appendix A), and optimization drives the policy towards a uniform distribution: if $\pi_{\boldsymbol{\theta}}(a|s) > \frac{1}{|\mathcal{A}|}$, the sign is negative, and the probability of $a$ is decreased; instead, if $\pi_{\boldsymbol{\theta}}(a|s) < \frac{1}{|\mathcal{A}|}$, the sign becomes positive, and the probability of $a$ is increased. Conversely, the gradient of the disequilibrium term is $\nabla_{\boldsymbol{\theta}}D[\pi_{\boldsymbol{\theta}}](s) = \sum_a 2(\pi_{\boldsymbol{\theta}}(a|s)-\frac{1}{|\mathcal{A}|})\nabla_{\boldsymbol{\theta}}\pi_{\boldsymbol{\theta}}(a|s)$, whose sign is positive when the action probability is greater than $\frac{1}{|\mathcal{A}|}$ and negative when it is lower, thus pushing the distribution even further from equilibrium. Finally, the gradient of the complexity bonus $C[\pi_{\boldsymbol{\theta}}](s) = S[\pi_{\boldsymbol{\theta}}](s) \cdot D[\pi_{\boldsymbol{\theta}}](s)$ is given by the product rule:

$$\nabla_{\boldsymbol{\theta}}C[\pi_{\boldsymbol{\theta}}](s) = \sum_a \left[-D[\pi_{\boldsymbol{\theta}}](s)(\log\pi_{\boldsymbol{\theta}}(a|s)+1) + 2S[\pi_{\boldsymbol{\theta}}](s)(\pi_{\boldsymbol{\theta}}(a|s)-\frac{1}{|\mathcal{A}|})\right]\nabla_{\boldsymbol{\theta}}\pi_{\boldsymbol{\theta}}(a|s). \quad (6)$$

The term in the brackets, which has $|\mathcal{A}|$ maxima (see Appendix A), dictates the update and has a less straightforward behavior that depends on the current policy. When the policy is near-deterministic, i.e., $S \approx 0$, the gradient is dominated by the first term, which incentivizes exploration. Conversely, when the policy is near-uniform, i.e., $D \approx 0$, the gradient is dominated by the second term, which encourages the policy to become more deterministic. Finally, when both entropy and disequilibrium are large ($S, D \gg 0$), both terms influence the gradient sign, creating a dynamic equilibrium that keeps the policy stochastic without collapsing into a simple, deterministic strategy or dissolving into pure randomness. Overall, this self-regulating mechanism, which encourages exploration when the policy is nearly deterministic and imposes structure when it approaches pure randomness, offers a theoretical explanation for the stability and robustness of CDPO observed in our experiments. This highlights the interplay between exploration and exploitation that enables consistent learning and prevents collapse into trivial or random policies.

## 5 EXPERIMENTS

### 5.1 EXPERIMENTAL SETUP

We evaluate CDPO on a diverse suite of environments with discrete, variable-length action space: CartPole (2-dimensional action space), CarRacing (5), and the Atari games AirRaid (6), Asteroids

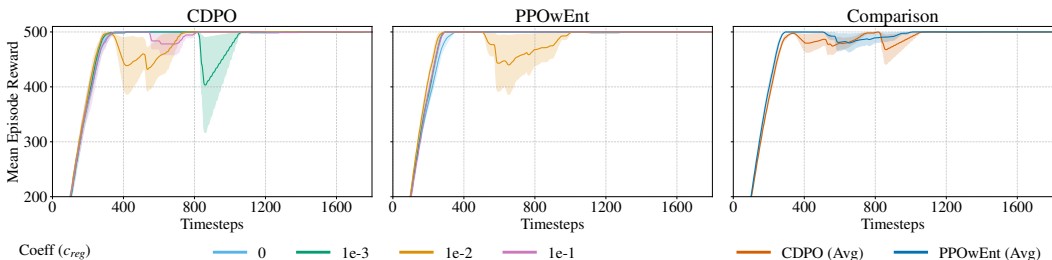

Figure 2: CartPole mean return for different $c_{reg}$ values of CDPO (left) and PPOwEnt (center), and their aggregated average (right). The mean and standard error are shown across 3 seeds.

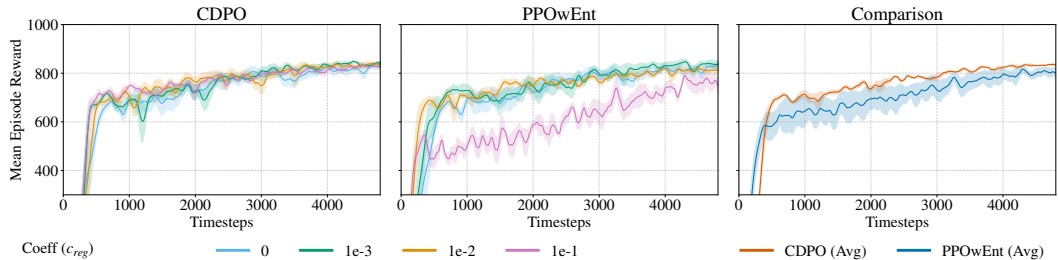

Figure 3: CarRacing mean return for different $c_{reg}$ values of CDPO (left) and PPOwEnt (center), and their aggregated average (right). The mean and standard error are shown across 3 seeds.

(14), and Riverraid (18), all taken from `gymnasium` (Towers et al., 2024); and CoinRun (15) from `procgen` (OpenAI, 2019).

We compare CDPO against two key baselines: i) PPO with an entropy bonus (from hereon, PPOwEnt), and ii) PPO without entropy (PPOwoEnt) to establish the floor performance and demonstrate the need for regularization in harder environments. For both CDPO and PPOwEnt, we vary the regularization coefficient $c_{reg}$ to evaluate their robustness. We test the values [1e-1, 1e-2, 1e-3] for the two simplest environments, and we extend them to [1e-1, 5e-2, 1e-2, 5e-3, 1e-3] for the others. We repeat all the experiments across 3 different seeds. The architectures and hyperparameters used are taken from `RL-Baselines3-Zoo` (Raffin et al., 2021) and reported in Appendix B.

## 5.2 EXPERIMENTAL RESULTS

The results of the experiments are reported in Figures 2 to 7. Each figure shows (a) the learning curves for CDPO and (b) entropy-regularized PPO under different coefficient values (including PPO without entropy), and also reports (c) a comparison of their aggregated performance across all non-zero coefficients. Our experiments reveal three distinct patterns regarding the role of the regularization term.

**Environments where regularization has minimal impact.** In simpler tasks like CartPole (Fig. 2) and CarRacing (Fig. 3), exploration does not impact the final results. Here, both CDPO and PPOwEnt perform on par with PPOwoEnt, demonstrating that the complexity bonus does not hinder performance when it is not needed. However, very high entropy coefficients can slightly slow down PPOwEnt learning, a sensitivity not observed with CDPO.

**Environments where entropy regularization is detrimental.** In environments like CoinRun (Fig. 4), aggressive exploration is counterproductive. This might be because only a small percentage of available actions is actually helpful; blind exploration forces the agent to select ineffective actions, practically preventing (fast) convergence. Indeed, increasing the entropy coefficient severely degrades PPO performance, preventing it from reaching the optimal solution. In contrast, CDPO remains robust across all coefficient values, consistently matching or improving upon the baseline performance by avoiding an overly random policy. A similar pattern is visible in AirRaid (Fig. 5),

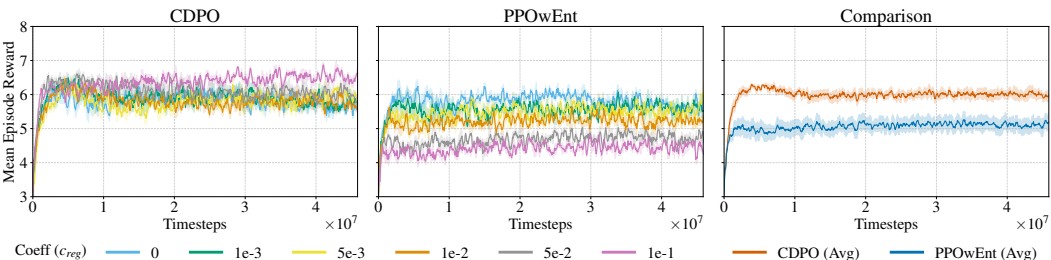

Figure 4: CoinRun mean return for different $c_{reg}$ values of CDPO (left) and PPOwEnt (center), and their aggregated average (right). The mean and standard error are shown across 3 seeds.

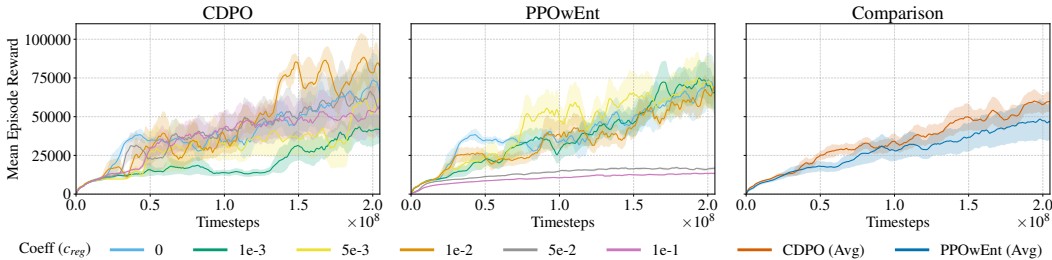

Figure 5: AirRaid mean return for different $c_{reg}$ values of CDPO (left) and PPOwEnt (center), and their aggregated average (right). The mean and standard error are shown across 3 seeds.

where high entropy coefficients completely stall learning, while CDPO maintains stability over a wider range of values.

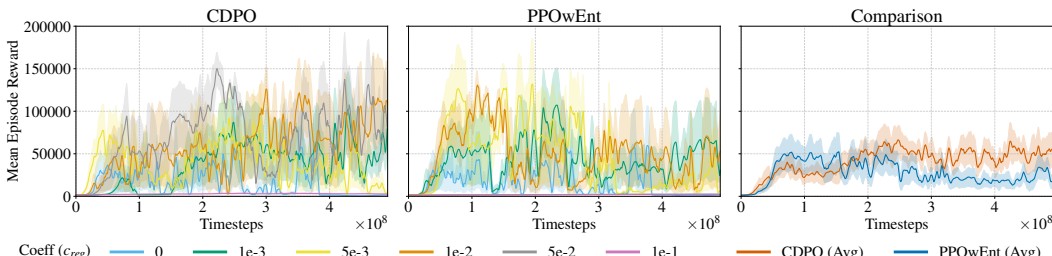

Figure 6: Asteroids mean return for different $c_{reg}$ values of CDPO (left) and PPOwEnt (center), and their aggregated average (right). The mean and standard error are shown across 3 seeds.

**Environments where regularization is beneficial.** For more complex tasks like Asteroids (Fig. 6) and RiverRaid (Fig. 7), effectively balancing exploration and exploitation is key to achieving high scores. Here, PPOwoEnt is sub-optimal or even incapable of maximizing the reward. While a carefully tuned entropy bonus can improve performance over the baseline, PPOwEnt is highly sensitive to the coefficient value $c_{reg}$: a too-high value deteriorates performance, while a too-low value is ineffective. CDPO achieves comparable or superior results to a well-tuned entropy bonus but does so across a much broader range of coefficients.

## 5.3 CARTERPILLAR

To systematically evaluate how regularization affects performance as task complexity increases, we designed a new environment with tunable difficulty. Standard benchmarks are often fixed, making it difficult to observe the varying necessity of regularization strategies. Our proposed environment,

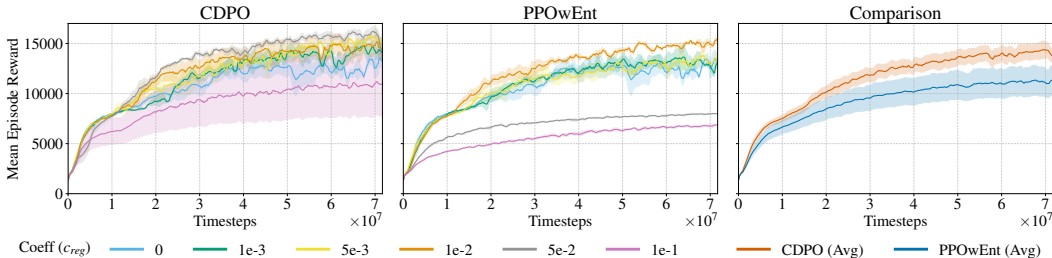

Figure 7: RiverRaid mean return for different $c_{reg}$ values of CDPO (left) and PPOwEnt (center), and their aggregated average (right). The mean and standard error are shown across 3 seeds.

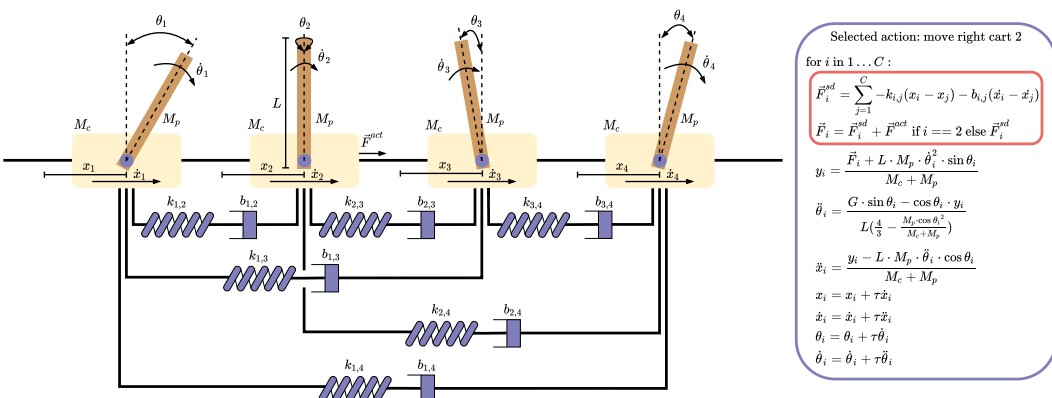

Figure 8: Schematic representation of the CARTerpillar environment with 4 carts. On the left, a flattened rendering with all the relevant physical symbols. On the right, the full dynamics of the environment, where the effect of newly introduced dampers and springs is highlighted in red.

which we name CARTerpillar[1], is an extension of the classic CartPole environment to multiple carts, where the number of carts $C$ to be balanced is a parameter that controls the difficulty of the environment. The $C$ carts form a fully-connected system, with all possible pairs of carts $i, j$ linked by a damper (with a constant factor of $k_{i,j} = k$) and a spring (with a constant factor of $b_{i,j} = b$). Adding an $i$-th cart increases the number of connections by $i - 1$, making the dynamics more *complex*, and expands the discrete action space by two actions (push left/right for the new cart). At each timestep, the agent can only move one cart left or right; however, each cart is also subject to a force that is due to all the dampers and springs connecting them. Figure 8 reports a simplified rendering of CARTerpillar, together with its full physical dynamics.

We compare CDPO against PPO with and without entropy regularization over CARTerpillar across different $C$ values. For $C < 9$, exploration is not strictly necessary: all methods converge to the optimum. Instead, for $C \in [9, 10, 11]$, the results aggregated over the five different $c_{reg}$ values (summarized in Figure 9) clearly demonstrate the relationship between task difficulty and the utility of regularization. As the number of carts increases, the performance of the baseline PPOwoEnt drops significantly, highlighting the growing need for structured exploration. While entropy regularization becomes beneficial for harder configurations (e.g., 10 and 11 carts), our complexity-based approach proves to be more robust to different $c_{reg}$ values (see Appendix C for a detailed analysis). To sum up, CDPO does not hinder performance in simpler settings and consistently outperforms PPOwEnt in the more challenging, high-dimensional scenarios.

---

[1]A `gym`-based implementation can be found at: `https://anonymous.4open.science/r/CARTerpillar/readme.md`

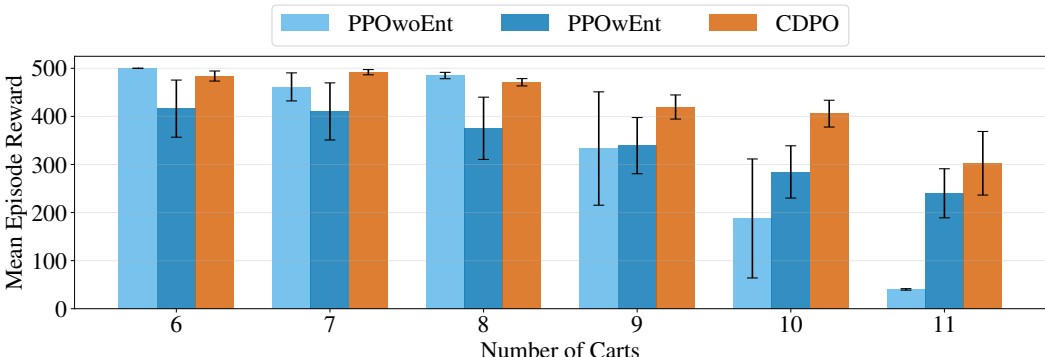

Figure 9: CARTerpillar aggregated results. Each bar reports the aggregated mean episode reward and the standard error across three seeds and all tested $c_{reg}$ coefficients.

## 6    DISCUSSION

Overall, our experimental results demonstrate that CDPO offers a more stable and reliable alternative to entropy regularization. It effectively adapts the level of exploration pressure, proving beneficial in complex environments while remaining harmless in simpler ones, thereby reducing the need for meticulous hyperparameter tuning. Specifically, CDPO encourages sufficient exploration when necessary, maintaining a stochastic (but not random) policy. When regularization is unnecessary, as in simpler environments, its inclusion does not negatively affect performance, and CDPO performs on par with the non-regularized PPO across all tested coefficients for the complexity term. This holds even in cases where regularization is harmful, such as when a policy must become occasionally deterministic to make precise decisions. Unlike entropy, complexity has a more nuanced landscape, where solutions with near-zero probabilities in certain dimensions can still lie near an optimum. In other words, a policy with many near-zero probability actions may have very low entropy, yet still retain high complexity. This distinction ensures that complexity-based regularization remains effective even in scenarios where entropy regularization degrades performance.

In summary, CDPO emerges as a strong regularization method, especially in settings where the appropriate level of exploration is unknown. Its robustness to the choice of $c_{reg}$ can drastically reduce the need for hyperparameter tuning, with substantial savings in energy consumption and computational cost, and can enable faster adaptation in dynamic or non-stationary environments where models may need frequent retraining or fine-tuning. However, the $c_{reg}$ coefficient still influences the final result, though less significantly than for entropy. Additionally, our evaluation is limited to classic RL scenarios with a relatively small action space. Whether the observed benefits scale up with the number of actions is an open question. Finally, due to the mathematical formulation of disequilibrium, CDPO currently applies only to environments with discrete action spaces. Extending our approach to continuous actions represents a compelling direction for future work.

## 7    CONCLUSION

Entropy regularization plays a key role in policy optimization algorithms like PPO by fostering exploration, yet its scaling factor is difficult to tune, and its push towards randomness can sometimes be detrimental in simpler settings. In this paper, we propose addressing these issues by replacing entropy with complexity, i.e., scaling entropy with the policy disequilibrium. While still preventing determinism in the policy, this bonus also avoids purely random behavior. We have conducted extensive experiments on classical reinforcement learning environments and CARTerpillar, a modified CartPole that allows for a fine-grained control of its difficulty, finding that, especially for harder tasks, complexity improves upon entropy regularization by being more resilient to its scaling factor and providing better results in more complex environments. Our research agenda includes extending the scope of complexity regularization to different policy gradient methods and to the maximum-entropy framework, and studying whether its adoption in more practical scenarios, such as language modeling and decision making, can further stabilize reinforcement learning algorithms.

ETHICS STATEMENT

We believe this paper does not raise specific ethical concerns, given the algorithmic nature of our contribution. Nevertheless, the authors are aware of the potential negative applications of these technologies in the future, and these considerations are always at the center of our research agenda.

REPRODUCIBILITY STATEMENT

To ensure the reproducibility of our work, we provide comprehensive details throughout the paper and its appendices. The formal definition of our proposed Complexity-Driven Policy Optimization (CDPO) algorithm and its pseudocode are presented in Section 4. Our experimental setup is outlined in Section 5, with full implementation details, including network architectures and all hyperparameters, available in Appendix B. The source code for all CDPO experiments is available as supplementary material at `https://anonymous.4open.science/r/CDPO/README.md`.

LARGE LANGUAGE MODEL USAGE STATEMENT

We acknowledge the use of large language models for the sole purpose of polishing writing, finding typos, and helping check the grammar correctness of the manuscript.

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

## A  PROOFS

Entropy is defined as $S[\pi_{\boldsymbol{\theta}}](s) = \sum_a -\pi_{\boldsymbol{\theta}}(a|s) \log \pi_{\boldsymbol{\theta}}(a|s)$ and subject to the probability constraint $\sum_a \pi_{\boldsymbol{\theta}}(a|s) = 1$. To compute its gradient subject to the constraint, we can make use of the method of Lagrange multipliers, where the gradient of $f(x)$ subject to $g(x) = 0$ can be found by defining the Lagrangian $\mathcal{L}(x, \lambda) = f(x) + \lambda g(x)$:

$$
\begin{aligned}
\mathcal{L}(\boldsymbol{\theta}, \lambda) &= \sum_a (-\pi_{\boldsymbol{\theta}}(a|s) \log \pi_{\boldsymbol{\theta}}(a|s)) + \lambda \sum_a (\pi_{\boldsymbol{\theta}}(a|s)) - 1 \\
&= \sum_a (-\pi_{\boldsymbol{\theta}}(a|s) \log \pi_{\boldsymbol{\theta}}(a|s) + \lambda \pi_{\boldsymbol{\theta}}(a|s)) - 1 \\
&= -\sum_a ((\log \pi_{\boldsymbol{\theta}}(a|s) - \lambda) \pi_{\boldsymbol{\theta}}(a|s)) - 1.
\end{aligned}
\tag{7}
$$

Now, we can compute its gradient with the product rule:

$$
\begin{aligned}
\nabla_{\boldsymbol{\theta}} \mathcal{L}(\boldsymbol{\theta}, \lambda) &= \nabla_{\boldsymbol{\theta}} \left[ -\sum_a (\log \pi_{\boldsymbol{\theta}}(a|s) - \lambda) \pi_{\boldsymbol{\theta}}(a|s) \right] - \nabla_{\boldsymbol{\theta}} 1 \\
&= -\sum_a (\log \pi_{\boldsymbol{\theta}}(a|s) - \lambda) \nabla_{\boldsymbol{\theta}} \pi_{\boldsymbol{\theta}}(a|s) + \pi_{\boldsymbol{\theta}}(a|s) \nabla_{\boldsymbol{\theta}} (\log \pi_{\boldsymbol{\theta}}(a|s) - \lambda) \\
&= -\sum_a (\log \pi_{\boldsymbol{\theta}}(a|s) - \lambda) \nabla_{\boldsymbol{\theta}} \pi_{\boldsymbol{\theta}}(a|s) + \pi_{\boldsymbol{\theta}}(a|s) \nabla_{\boldsymbol{\theta}} \log \pi_{\boldsymbol{\theta}}(a|s) - \pi_{\boldsymbol{\theta}}(a|s) \nabla_{\boldsymbol{\theta}} \lambda,
\end{aligned}
\tag{8}
$$

where the last term is null. Given that $\nabla \log \pi(a|s) = \frac{\nabla \pi(a|s)}{\pi(a|s)}$, we can replace $\nabla_{\boldsymbol{\theta}} \log \pi(a|s)$ in the second term, obtaining:

$$
\begin{aligned}
\nabla_{\boldsymbol{\theta}} \mathcal{L}(\boldsymbol{\theta}, \lambda) &= -\sum_a (\log \pi_{\boldsymbol{\theta}}(a|s) - \lambda) \nabla_{\boldsymbol{\theta}} \pi_{\boldsymbol{\theta}}(a|s) + \pi_{\boldsymbol{\theta}}(a|s) \frac{\nabla_{\boldsymbol{\theta}} \pi_{\boldsymbol{\theta}}(a|s)}{\pi_{\boldsymbol{\theta}}(a|s)} \\
&= -\sum_a (\log \pi_{\boldsymbol{\theta}}(a|s) - \lambda) \nabla_{\boldsymbol{\theta}} \pi_{\boldsymbol{\theta}}(a|s) + \nabla_{\boldsymbol{\theta}} \pi_{\boldsymbol{\theta}}(a|s) \\
&= -\sum_a (\log \pi_{\boldsymbol{\theta}}(a|s) + 1 - \lambda) \nabla_{\boldsymbol{\theta}} \pi_{\boldsymbol{\theta}}(a|s).
\end{aligned}
\tag{9}
$$

Since the aim is to maximize entropy, we need to find where all partial derivatives are equal to 0, i.e., where the original constraint $g(x) = \sum_a (\pi_{\boldsymbol{\theta}}(a|s)) - 1 = 0$ is satisfied, and when $\log \pi_{\boldsymbol{\theta}}(a|s) + 1 - \lambda = 0, \forall a \in \mathcal{A}$. The last set of equations shows that all $\pi_{\boldsymbol{\theta}}(a|s)$ must be equal; under the constraint, this means that $\pi_{\boldsymbol{\theta}}(a|s) = \frac{1}{|\mathcal{A}|}, \forall a \in \mathcal{A}$, and that $\lambda = 1 + \log \frac{1}{|\mathcal{A}|} = 1 + \log 1 - \log |\mathcal{A}| = 1 - \log |\mathcal{A}|$. When $\log \pi_{\boldsymbol{\theta}}(a|s) < \lambda - 1$, i.e., $\pi_{\boldsymbol{\theta}}(a|s) < \frac{1}{|\mathcal{A}|}$, the sign of the gradient is positive, and the probability gets increased; vice versa, when $\log \pi_{\boldsymbol{\theta}}(a|s) > \lambda - 1$, the sign of the gradient becomes negative, and the probability is decreased.

Disequilibrium is defined as $D[\pi_{\boldsymbol{\theta}}](s) = \sum_a (\pi_{\boldsymbol{\theta}}(a|s) - \frac{1}{|\mathcal{A}|})^2$ and subject to the probability constraint $\sum_a \pi_{\boldsymbol{\theta}}(a|s) = 1$. Again, we can make use of the method of the Lagrange multipliers, obtaining:

$$\mathcal{L}(\boldsymbol{\theta}, \lambda) = \sum_a \left( \pi_{\boldsymbol{\theta}}(a|s) - \frac{1}{|\mathcal{A}|} \right)^2 + \lambda \left( \sum_a \pi_{\boldsymbol{\theta}}(a|s) - 1 \right)$$

$$= \sum_a \left( \pi_{\boldsymbol{\theta}}^2(a|s) + \frac{1}{|\mathcal{A}|^2} - \frac{2}{|\mathcal{A}|} \pi_{\boldsymbol{\theta}}(a|s) \right) + \lambda \sum_a \pi_{\boldsymbol{\theta}}(a|s) - \lambda$$

$$= \sum_a \left( \pi_{\boldsymbol{\theta}}^2(a|s) - \frac{2}{|\mathcal{A}|} \pi_{\boldsymbol{\theta}}(a|s) + \lambda \pi_{\boldsymbol{\theta}}(a|s) \right) + \frac{1}{|\mathcal{A}|} - \lambda \qquad (10)$$

$$= \sum_a \left( \pi_{\boldsymbol{\theta}}^2(a|s) + \frac{\lambda|\mathcal{A}| - 2}{|\mathcal{A}|} \pi_{\boldsymbol{\theta}}(a|s) \right) + \frac{1}{|\mathcal{A}|} - \lambda.$$

Its gradient with respect to $\theta$ can be defined as follows:

$$\nabla_{\boldsymbol{\theta}} \mathcal{L}(\boldsymbol{\theta}, \lambda) = \sum_a \left( 2\pi_{\boldsymbol{\theta}}(a|s) \nabla_{\boldsymbol{\theta}} \pi_{\boldsymbol{\theta}}(a|s) + \frac{\lambda|\mathcal{A}| - 2}{|\mathcal{A}|} \nabla_{\boldsymbol{\theta}} \pi_{\boldsymbol{\theta}}(a|s) \right)$$

$$= \sum_a \left( 2\pi_{\boldsymbol{\theta}}(a|s) + \frac{\lambda|\mathcal{A}| - 2}{|\mathcal{A}|} \right) \nabla_{\boldsymbol{\theta}} \pi_{\boldsymbol{\theta}}(a|s). \qquad (11)$$

In order to find the optimum, we need to set all partial derivatives equal to 0, including with respect to $\lambda$, i.e., that the original constraint $g(x) = \sum_a (\pi_{\boldsymbol{\theta}}(a|s)) - 1 = 0$ is satisfied, plus that $2\pi_{\boldsymbol{\theta}}(a|s) + \frac{\lambda|\mathcal{A}|-2}{|\mathcal{A}|} = 0, \forall a \in \mathcal{A}$. These equations are true for $\pi_{\boldsymbol{\theta}}(a|s) = \frac{1}{|\mathcal{A}|} - \frac{\lambda}{2}$. But since we also have the original constraint, $\sum_a \frac{1}{|\mathcal{A}|} - \frac{\lambda}{2} - 1 = 1 - \frac{\lambda|\mathcal{A}|}{2} - 1 = -\frac{\lambda|\mathcal{A}|}{2}$ must be equal to zero; thus, $\lambda = 0$ and $\pi_{\boldsymbol{\theta}}(a|s) = \frac{1}{|\mathcal{A}|}, \forall a \in \mathcal{A}$. As expected, the minimum lies where the distribution is equiprobable. When $\pi_{\boldsymbol{\theta}}(a|s) < \frac{1}{|\mathcal{A}|}$, the gradient is negative, causing the probability of selecting action $a$ to decrease further. Conversely, when $\pi_{\boldsymbol{\theta}}(a|s) > \frac{1}{|\mathcal{A}|}$, the gradient becomes positive, leading to an increase in the probability of selecting action $a$.

Finally, complexity is defined as $C[\pi_{\boldsymbol{\theta}}](s) = D[\pi_{\boldsymbol{\theta}}](s)S[\pi_{\boldsymbol{\theta}}](s)$. According to the product rule, its gradient is as follows:

$$\nabla_{\boldsymbol{\theta}} C[\pi_{\boldsymbol{\theta}}](s) = S[\pi_{\boldsymbol{\theta}}](s) \nabla_{\boldsymbol{\theta}} D[\pi_{\boldsymbol{\theta}}](s) + D[\pi_{\boldsymbol{\theta}}](s) \nabla_{\boldsymbol{\theta}} S[\pi_{\boldsymbol{\theta}}](s)$$

$$= \sum_a \left[ 2S[\pi_{\boldsymbol{\theta}}](s) \left( \pi_{\boldsymbol{\theta}}(a|s) - \frac{1}{|\mathcal{A}|} \right) - D[\pi_{\boldsymbol{\theta}}](s) \left( \log \pi_{\boldsymbol{\theta}}(a|s) + 1 \right) \right] \nabla_{\boldsymbol{\theta}} \pi_{\boldsymbol{\theta}}(a|s)$$

$$= \sum_a \left[ 2S[\pi_{\boldsymbol{\theta}}](s)\pi_{\boldsymbol{\theta}}(a|s) - D[\pi_{\boldsymbol{\theta}}](s) \log \pi_{\boldsymbol{\theta}}(a|s) - \frac{2}{|\mathcal{A}|} S[\pi_{\boldsymbol{\theta}}](s) - D[\pi_{\boldsymbol{\theta}}](s) \right] \nabla_{\boldsymbol{\theta}} \pi_{\boldsymbol{\theta}}(a|s).$$
$$(12)$$

However, as for entropy and disequilibrium, complexity is subject to the probability constraint $\sum_a \pi_{\boldsymbol{\theta}}(a|s) = 1$ as well. To find the local optima of complexity, we need to find where the partial derivatives of the Lagrangian $C[\pi_{\boldsymbol{\theta}}](s) - \lambda(\sum_a \pi_{\boldsymbol{\theta}}(a|s) - 1)$ are equal to 0, i.e., to solve the equation system composed of the probability constraint plus $2S[\pi_{\boldsymbol{\theta}}](s)\pi_{\boldsymbol{\theta}}(a|s) - D[\pi_{\boldsymbol{\theta}}](s) \log \pi_{\boldsymbol{\theta}}(a|s) - \frac{2}{|\mathcal{A}|}S[\pi_{\boldsymbol{\theta}}](s) - D[\pi_{\boldsymbol{\theta}}](s) - \lambda = 0, \forall a \in \mathcal{A}$. This system has multiple solutions, i.e., where $\pi_{\boldsymbol{\theta}}(a|s) = \frac{1}{|\mathcal{A}|}, \forall a \in \mathcal{A}$ (which represents the local minimum and complexity is equal to 0), plus $|\mathcal{A}|$ maxima that are the recombination of the same set of probabilities.

## B  IMPLEMENTATION DETAILS

The experiments were carried out on a system equipped with a 32-core AMD EPYC 7413 processor and an NVIDIA L40 GPU, running Python version 3.10.18. We used the `torch` library and

leveraged the `stable-baselines3` implementation of PPO, which was modified into CDPO according to Algorithm B.

---

**Python implementation of CDPO complexity loss**

```
...
dist = self.policy.get_distribution(rollout_data.observations)
probs = dist.distribution.probs
entropy = dist.distribution.entropy()
disequilibrium = torch.sum((probs − 1.0/probs.size(−1))**2, dim = −1)
complexity = entropy * disequilibrium
complexity_loss = −torch.mean(complexity)
...
```

---

The adopted neural networks followed the default architectures provided by `stable-baselines3`. For 1D observation space environments, i.e., CartPole and CARTerpillar environments, we used an MLP with 2 fully connected layers with 64 units each. For image observation spaces, i.e., CarRacing and Atari environments, we used a Nature CNN architecture for feature extraction, with a sequence of 2D convolutional layers with layer sizes $[32, 64, 64]$, kernel sizes $[(8, 8), (4, 4), (3, 3)]$, and strides $[(4, 4), (2, 2), (1, 1)]$, with ReLU activation function. The output is then flattened and passed through a dense layer (of dimension 256 for CarRacing and 512 for Atari games) with ReLU activation. Finally, consistent with the original CoinRun experiments (Cobbe et al., 2020), we used the IMPALA CNN architecture (Espeholt et al., 2018) with three layers, each consisting of a 2D convolutional layer, a 2D max-pooling operation with kernel size 3 and stride 2, followed by two residual blocks. Each residual block contains two 2D convolutional layers preceded by ReLU activation functions, and their output is summed with their input. All the convolutional layers inside the three IMPALA layers have the same size ($[16, 32, 32]$, respectively), and the same stride of 3. Finally, the output is flattened and passed through a dense layer of dimension 256 with ReLU activation.

The full list of training hyperparameters is reported in Table 1.

| Parameter | CartPole | CarRacing | CoinRun | Atari games | CARTerpillar |
|---|---|---|---|---|---|
| Number of parallel envs | 8 | 8 | 256 | 8 | 8 |
| Steps between updates | 32 | 512 | 256 | 128 | 32 |
| Policy epochs | 20 | 20 | 3 | 4 | 20 |
| Batch size | 256 | 128 | 2048 | 256 | 256 |
| GAE lambda | 0.8 | 0.95 | 0.95 | 0.95 | 0.8 |
| Gamma | 0.98 | 0.99 | 0.99 | 0.99 | 0.98 |
| Max gradient norm | 0.5 | 0.5 | 0.5 | 0.5 | 0.5 |
| Learning rate | 1e-3 | 1e-4 | 5e-4 | 2.5e-4 | 1e-3 |
| Clip range $\epsilon$ | 0.2 | 0.2 | 0.2 | 0.1 | 0.2 |
| Value-function coeff. $c_{vf}$ | 0.5 | 0.5 | 0.5 | 0.5 | 0.5 |
| Frame skip | - | 2 | - | - | - |
| Frame stack | - | 2 | - | 4 | - |
| SDE sample frequency | - | 4 | - | - | - |
| Number of parallel workers | - | - | 2 | - | - |
| Number of training levels | - | - | 500 | - | - |
| Number of testing levels | - | - | $\infty$ | - | - |
| Number of minibatches | - | - | 8 | - | - |
| Distribution mode | - | - | Hard | - | - |
| Value function clipping | - | - | True | - | - |
| Max episode steps | - | - | - | - | 500 |
| Gravity constant $G$ | - | - | - | - | 9.81 |
| Spring constant $k$ | - | - | - | - | 1 |
| Damper constant $b$ | - | - | - | - | 1 |

Table 1: Training parameters for all the experiments.

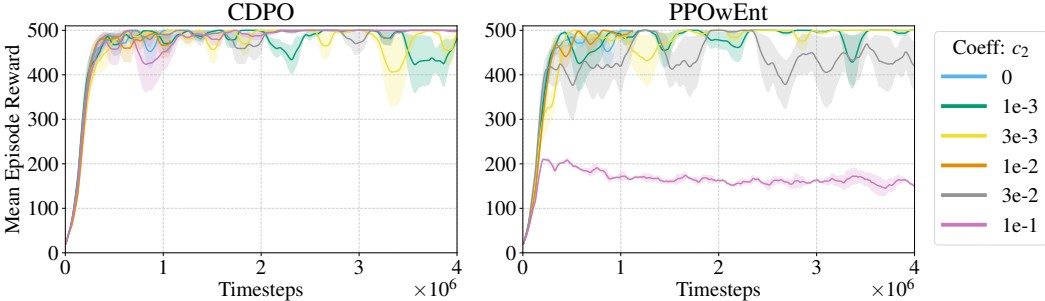

Figure 10: Mean episode reward for CARTerpillar with $C = 6$ carts for CDPO (left) and PPOwEnt (right) with different $c_{reg}$ values. Each plot reports the mean and standard error over 3 seeds.

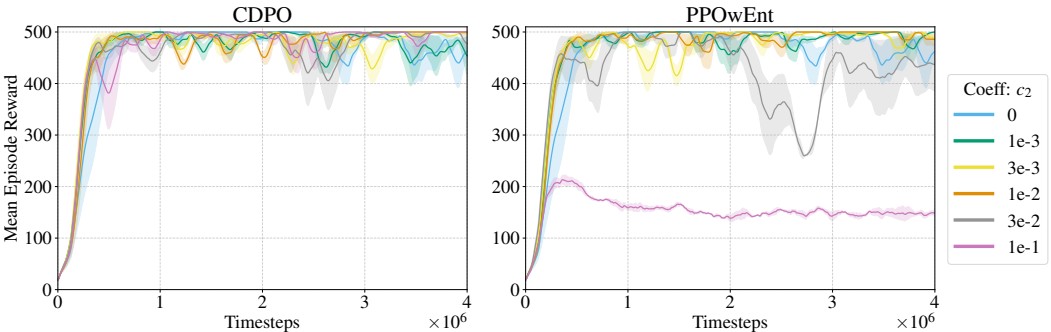

Figure 11: Mean episode reward for CARTerpillar with $C = 7$ carts for CDPO (left) and PPOwEnt (right) with different $c_{reg}$ values. Each plot reports the mean and standard error over 3 seeds.

## C CARTERPILLAR ADDITIONAL RESULTS

In the following, we report the detailed results of CDPO and PPOwEnt for different CARTerpillar configurations. In particular, we evaluate their performances for five $c_{reg}$ values: [1e-1, 3e-2, 1e-2, 3e-3, 1e-3]. For 6, 7, and 8 carts (Figures 10, 11, and 12, respectively), all $c_{reg}$ values reach the optimum in the case of CDPO, while some of them struggle in the case of PPOwEnt. The same holds for 9 carts (Figure 13), where a too-high entropy coefficient can make the agent unlearn optimal strategies. Finally, only a well-tuned coefficient can make PPOwEnt converge for 10 and 11 carts (Figures 14 and 15), while CDPO is able to converge with multiple $c_{reg}$ values. This greater robustness to variations in $c_{reg}$ is demonstrated in Figure 16, which presents the mean and standard error of the aggregated results for all evaluated cart numbers.

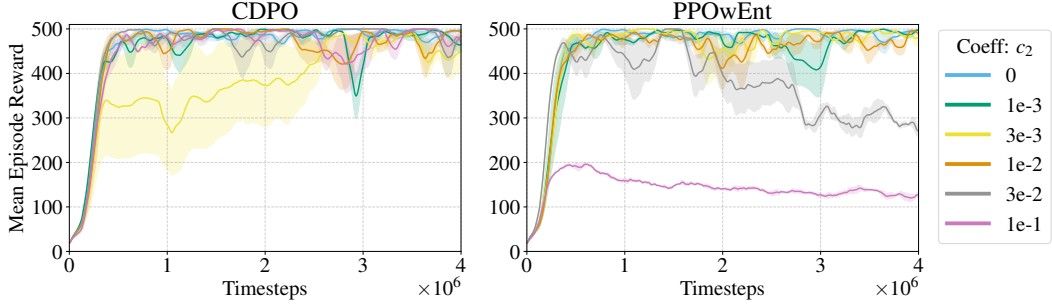

Figure 12: Mean episode reward for CARTerpillar with $C = 8$ carts for CDPO (left) and PPOwEnt (right) with different $c_{reg}$ values. Each plot reports the mean and standard error over 3 seeds.

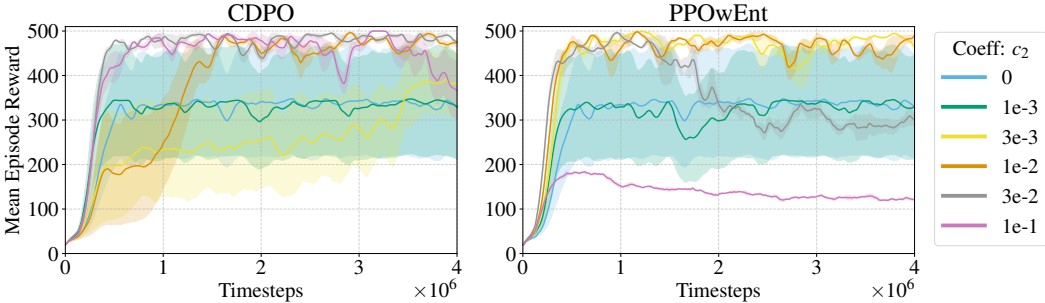

Figure 13: Mean episode reward for CARTerpillar with $C = 9$ carts for CDPO (left) and PPOwEnt (right) with different $c_{reg}$ values. Each plot reports the mean and standard error over 3 seeds.

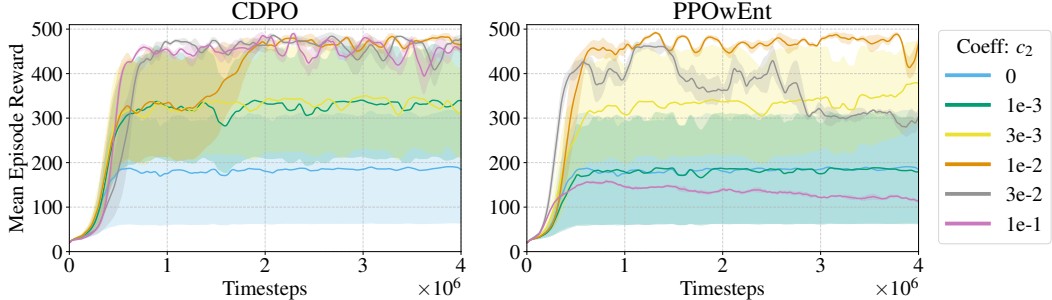

Figure 14: Mean episode reward for CARTerpillar with $C = 10$ carts for CDPO (left) and PPOwEnt (right) with different $c_{reg}$ values. Each plot reports the mean and standard error over 3 seeds.

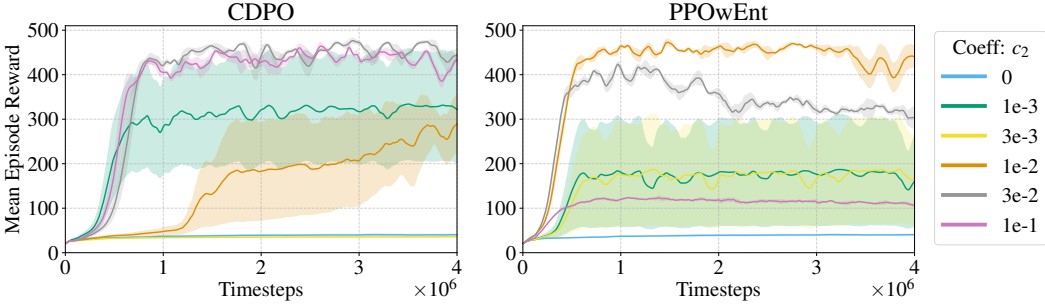

Figure 15: Mean episode reward for CARTerpillar with $C = 11$ carts for CDPO (left) and PPOwEnt (right) with different $c_{reg}$ values. Each plot reports the mean and standard error over 3 seeds.

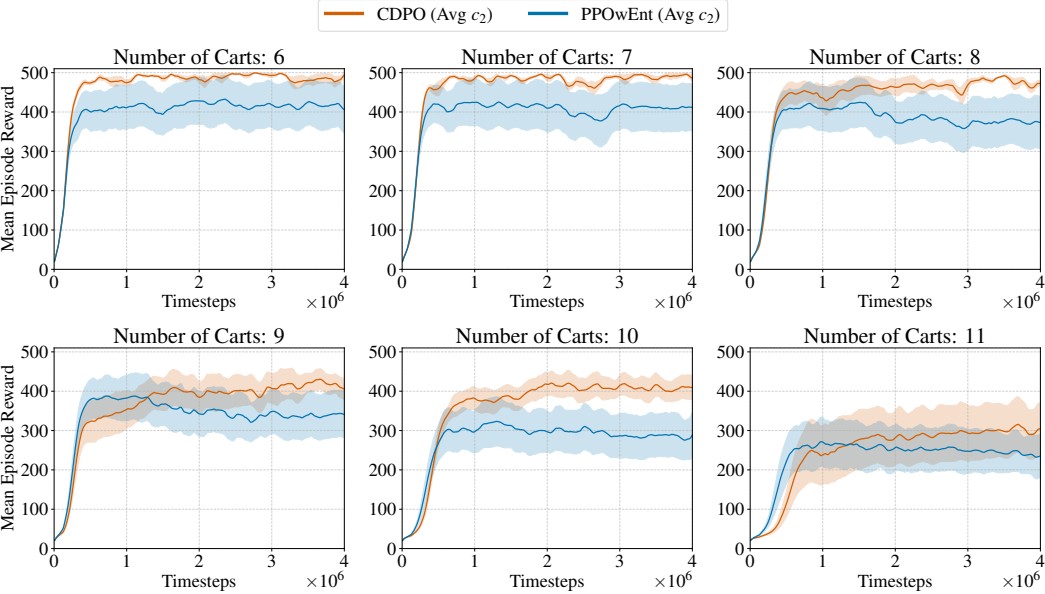

Figure 16: Aggregated results for CDPO and PPOwEnt over different $c_{reg}$ values for the CARTerpillar environment with varying number of carts. Each plot reports the mean and standard error over 3 seeds.

