# OpenReview forum: "Complexity-Driven Policy Optimization"
_ICLR.cc/2026/Conference — Submitted to ICLR 2026_

### Official Review · Reviewer_M85N · 2025-10-27

**Soundness:** 3
**Presentation:** 3
**Contribution:** 1
**Rating:** 2
**Confidence:** 4

**Summary:**

The authors present a novel use of LMC complexity as a regulariser for on-policy RL algorithms that helps with exploration in a more structured way than naive policy entropy regularisation. They argue LMC complexity favours policies that are neither fully deterministic nor fully uniform. They present a set of empirical results comparing against PPO with and without policy regularisation and show some improvements in terms of stability and rewards.

**Strengths:**

- The problem of how to efficiently and effectively induce exploration in RL is an extremely relevant one.
- The method is well presented and the intuition behind it is clear.
- The paper is well written.

**Weaknesses:**

- I am not convinced of the depth of the contribution. While the problem addressed is very relevant, I am not sure the modification proposed (using LMC as a regulariser) is enough of a contribution in itself: it is a (arguably small) modification over well studied entropy regularisers. The theoretical analysis is also quite limited, so that does not help understand the advantages of using LMC versus other existing methods for encouraging exploration. While I agree that simply regularising for fixed entropy penalties seems naive and inefficient, it has advantages: It is very interpretable (in terms of the effects it causes in the policies and in the optimisation problem solution) and well studied (also in e.g. off-policy RL) both theoretically [1] and empirically. The authors themselves admit that the behaviour of the gradient descent algorithm is quite complex in terms of the LMC gradients, increasing or decreasing action probabilities depending on the policy.
- For such a paper to have enough of a contribution to the field, (if lacking a theoretical one) I would expect a very thorough empirical study on the impact of LMC regularisers in a wide range of RL benchmarks. The evaluation is very limited, with only a handful of environments and only against PPO. While I understand why the authors would prefer to focus on a narrower empirical setting, it fails to add to the contribution. In general, I cannot consider the improvements observed to be of sufficient relevance. In most of the environments the improvement is only observed against some of the regulariser coefficients, and the CARTerpillar environment (while maybe illustrative) is quite constructed and not enough of an argument in itself.


[1] Levine, Sergey. "Reinforcement learning and control as probabilistic inference: Tutorial and review." arXiv preprint arXiv:1805.00909 (2018).

**Questions:**

- What is the reason behind comparing both regularisers with the same set of coefficients? Taking a Lagrangian interpretation, the regularisation coefficient fixes the optimisation constraint bound, but given that the two regularisers are fundamentally different there is no reason to believe they should be regularised with the same strength. It could very well be that the range picked is simply too wide, and it impacts the entropy regulariser more than the LMC.
- This raises a secondary point; most experimental results are reporter by averaging across all results for different coefficients, but if one takes the best performing one, the performance increase is not really apparent.

---

> ### Author Response · Authors · 2025-11-21
>
> We thank the reviewer for the insightful comments. Please find our responses below.
>
> ### Weaknesses
>
> 1) Although the substitution of entropy with complexity might appear as a small modification, its conceptual impact is substantial, as it adapts entropy regularization based on the determinism of the current policy. We believe this represents a meaningful and original contribution. To the best of our knowledge, no other entropy regularizers follow a similar intuition, as they typically rely on fixed or bounded multipliers. While entropy retains the advantage of interpretability, complexity offers a powerful and flexible approach that provides significant practical benefits. The simplicity of entropy is indeed advantageous because its intuitiveness allows users to adjust its behavior via the scaling factor (for example, if a policy becomes too deterministic, the entropy coefficient can be increased; if it never converges, the coefficient can be decreased). While such manual tuning would not be feasible with complexity, our results show that it is largely unnecessary, as complexity is more robust to the choice of its scaling factor, essentially trading some interpretability for greater reliability. Finally, we thank the reviewer for pointing us to that paper, which we were not previously aware of. It will help us strengthen the theoretical foundations of our work.
>
> 2) While we understand the reviewer’s perspective, we believe that the six environments we considered (seven, including CARTerpillar) already cover a wide range of scenarios, exhibiting different properties in terms of action space size and requiring varying levels of entropy regularization. The CARTerpillar environment was specifically introduced to complement this analysis and to illustrate the relationship between the required regularization and the complexity of environmental dynamics. Although improvements are observed primarily for certain scaling factors, this highlights one of the main points of our findings: a carefully tuned complexity performs on par with a carefully tuned entropy, but when not finely tuned, complexity offers much stronger empirical guarantees.
>
> ### Questions
>
> 1) We completely agree on this point. However, the fact that complexity is less sensitive than entropy to the range of values is one reason for its greater robustness with respect to the scaling factor. We recognize that the two regularizers are different, and we selected a sufficiently broad range of values to encompass both optimal regions. While it is true that the optimal values for the entropy regularizer are generally smaller than those for the complexity regularizer, these optimal values also vary across environments. Importantly, complexity performs nearly optimally even when the chosen value is not the exact optimum.
>
> 2) We reported results both for individual hyperparameter settings and in aggregate for all experiments to illustrate this point: while optimal values are generally similar, non-optimal values for entropy perform worse than non-optimal values for complexity, highlighting the latter’s greater robustness and reliability in unknown scenarios. We acknowledge that this concept should have been emphasized more clearly throughout the paper.

---

> > ### Comment · Reviewer_M85N · 2025-11-25
> >
> > I appreciate the authors nuances and the discussion.
> > I believe that this paper could, in an optimal form, be an interesting contribution to the subfield of 'entropy/complexity measure'- regularised RL. However I still believe this should include a deep theoretical evaluation, or thorough empirical comparisons (not for the sake of simply adding benchmarks, but to provide a wide set of examples and intuitions over how LMC compares to other regularisers), or both. Thus, most of my concerns still stand, and I cannot fairly increase the score of my review.

---

### Official Review · Reviewer_fukB · 2025-10-31

**Soundness:** 2
**Presentation:** 2
**Contribution:** 2
**Rating:** 2
**Confidence:** 3

**Summary:**

The authors observe that maximum entropy methods can sometimes lead to excessive exploration. They propose an exploitation objective that mitigates this problem. The objective consists of two parts, one that motivates a uniform policy and the other a deterministic policy, thus allowing control over its randomness.

**Strengths:**

The overall observation is accurate and the approach is fair, supported by a method from the literature.

**Weaknesses:**

Several weaknesses limit the impact of the paper at this stage.
1. Some explanations and justifications are rather vague when it comes to the concept of exploration and, above all, structure. This is more a matter of presentation.
2. The literature on exploration methods is fairly limited. It only presents the foundation papers. In particular, much of the literature has already addressed some of the limitations of action-entropy exploration by introducing regularization on the visitation measure of states, state-actions, or features. This is discussed very briefly and is not compared to the proposed method empirically.
3. Section 2.2 is supposed to discuss the notion of complexity in RL but is limited to two papers. I find the discussion in this section lacking in substance.
4. The entropy function $S$ is not a function of state, as indicated in Eq. (2) and Eq. (5). It is defined in the pseudo-code, and the definition includes the expectation $E_t$, which marginalizes the effect of states.
5. Based on the pseudocode, I get the impression that only one iteration is performed on the PPO loss. This could be clarified. If this is the case, we end up with a classic policy gradient. Line 121, the motivation for PPO is slightly incorrect to me. In its initial version, PPO does not use a critic and does not really address the problem of policy gradient variance. The main advantage of PPO is that it reuses samples to perform several policy updates. The clip implements the idea of a “trust region”: if the policy has changed too much, the gradient is zero.
6. The regularization only works for discrete action spaces; this should be stated explicitly.
7. Section 4.2 is, in my opinion, incorrect. First, the existence of multiple optima does not imply anything about robustness. Furthermore, the discussion neglects the effect of expectation on states and does not discuss the optima as a function of theta. I do not think this section provides real insight, and certainly no theoretical justification for the superiority of the exploration objective.
8. In my opinion, the experimental setting does not allow for a proper comparison of exploration bonuses. Authors compare the aggregate effect over different weights on the regularization term. The performance for the average weight does not indicate how the two methods perform when they are correctly calibrated individually for the application. Typically, we observe here that PPOwEnt performs similarly to CDPO when we take the two best run for both methods. Smaller weigths are systematically needed for action-entropy regularization, which makes sense intuitively as it explores more agressively.
9. Benchmarks are so simple that the trivial method without exploration achieves already very good performance. Anyway, the difference between performance is very small.

**Questions:**

How is the gradient of the entropy bonus computed in practice?

---

> ### Author Response · Authors · 2025-11-21
>
> We thank the reviewer for the extensive comments. Please find the answers to all the concerns below.
>
> 1) We agree that these two concepts should have been presented in greater depth, as they form a core part of our research.
>
> 2) We realize that our exposition may have led to some misunderstandings. Our main goal is not to propose an exploration strategy, but rather a regularization strategy. While regularizing learning can influence exploration, its purpose is somewhat different: exploration strategies explicitly aim to gather informative knowledge about the space of solutions, whereas regularization strategies aim to prevent certain learning patterns, such as sub-optimal convergence or overly deterministic behaviors. This indirectly encourages a limited, more blind form of exploration, since avoiding determinism allows the policy to consider multiple options. Importantly, regularization techniques, particularly those affecting the policy at the loss level, are orthogonal to exploration strategies, which typically act at the reward level by introducing additional intrinsic signals. For this reason, we focused only on entropy regularization (and no regularization at all) as baselines in our experiments. We acknowledge that this was not sufficiently clear, and we plan to emphasize this central intuition more clearly in a revised version of the paper.
>
> 3) To the best of our knowledge, these were the only papers discussing RL and complexity together. We considered it important to include them, as otherwise our related work section would have completely lacked coverage of RL and complexity. If there are other relevant contributions that we may have overlooked, we would greatly appreciate it if the reviewer could point them out so that we can properly discuss them in the next version of the paper.
>
> 4) We respectfully disagree on this point. The entropy function depends on the state, since it is a function of the policy, which in turn depends on the state. However, we have now identified an error in the pseudocode: the mention of $s_t$ should not have been included, due to the presence of the expectation over multiple timesteps.
>
> 5) The fourth row of the pseudocode explicitly shows how we iterate over the same batch of collected trajectories across multiple policy epochs: “Loop for each policy epoch = 1 … N”. Regarding PPO, we did not refer to the use of a critic, but rather to a state-value function approximator. However, we agree that its introduction is not sufficiently clear and does not highlight its main contribution. We will revise this point accordingly.
>
> 6) We defined our problem in terms of discrete action spaces and noted this in the Discussion section. However, we agree that it should have been clearly stated in the Method section as well.
>
> 7) We understand that Section 4.2 can be misinterpreted on this matter. We did not want to imply that having multiple optima is necessarily a good thing in terms of robustness; the robustness comes from the fact that maximizing the complexity gradient lowers the probability of deterministic and purely random behaviors at the same time, instead of merely aiming for more randomness. Again, our aim was not to propose an exploration strategy, but rather a more effective regularization strategy (we mentioned exploration only because it affects the exploration-exploitation balance, which we consider beneficial). We would appreciate it if the reviewer could clarify why the relevance of states is emphasized here, as we are currently unsure how to address this point.
>
> 8) We did not report only the aggregated results, as is evident from all the figures in the Experiments section. Nevertheless, we provide them alongside the non-aggregated results because CDPO demonstrates greater robustness across different scaling factors. We are fully aware (and have never tried to obscure) that a carefully tuned entropy performs on par with a carefully tuned complexity. However, as noted in the Discussion section, complexity generally performs comparably to a carefully tuned entropy even when it is not finely tuned. In our view, this represents a significant advantage, as tuning the entropy multiplier is neither straightforward nor inexpensive.

---

> ### Author Response · Authors · 2025-11-21
>
> [continuing from previous comment...]
>
> 9) We chose benchmarks that were as different as possible from each other in terms of action space size and need for entropy regularization, so as to provide a clearer understanding of CDPO performances in scenarios where entropy may or may not be required. However, as our goal was to compare its performance in terms of robustness, we had to stick with environments that we know could be solved by PPO with or without entropy. This led to the selection of simpler environments.
>
> 10) The gradient of the entropy bonus is computed by adding the entropy bonus to the loss function; the gradient is automatically calculated and then backpropagated. As far as the computation of the entropy itself is considered, we used the `stable_baselines3` implementation of PPO, which uses the `entropy()` method from `torch.distributions.Categorical` to compute the entropy of the probability distribution (that is actually implemented as the sum of the -p*logp).

---

### Official Review · Reviewer_bMDS · 2025-11-01

**Soundness:** 3
**Presentation:** 3
**Contribution:** 2
**Rating:** 2
**Confidence:** 2

**Summary:**

This paper proposes a new regularization method for policy gradient algorithms, named Complexity-Driven Policy Optimization. The core idea is to replace the standard entropy bonus, which is often used to encourage exploration but can lead to inefficient policies, with a LMC complexity bonus.

**Strengths:**

- The paper identifies a good but overlooked problem with entropy regularization. Its objective is a purely random policy, which is not an optimal exploration strategy.
- The proposed solution of using LMC complexity to find a "structured stochastic" policy is simple and intuitive.

**Weaknesses:**

- I am not quite convinced of both the problem and the proposed solution. There are many easier ways to reduce the effect of regularization instead of the solution proposed in this work. The major and possibly only advantage of the proposed “complexity” measure is to balance stochasticity with structure. However, this complexity might also cause the policy to be neither structured nor explored enough. Since this new complexity measure is heuristic, the paper focuses on empirical experiments instead of theoretical analysis. However, the experiments conducted are not convincing. The paper only compares CDPO to entropy regularization. While this is the most direct comparison, it misses comparisons to the work relative to the broader field of exploration.
- Applicability limited to discrete action spaces.

**Questions:**

How does this policy-based regularization approach compare to reward-based intrinsic motivation methods (e.g., ICM, RND)? It's unclear if CDPO is an alternative to these methods or a complementary technique that could be combined with them.

---

> ### Author Response · Authors · 2025-11-21
>
> We thank the reviewer for pointing out these issues. We now realize that the scope of our work was not sufficiently clear. As correctly noted by the reviewer, our aim is to provide a technique for regularization, not for exploration. Accordingly, it is not an alternative to intrinsic motivation strategies or other curiosity-based RL methods designed to enhance exploration; rather, it is a regularization strategy that is orthogonal to and complementary with exploration strategies. We acknowledge that this was not sufficiently clear in our paper. For this reason, we only provided an experimental comparison with entropy regularization (and without any regularization).
>
> We agree that more attention to theoretical analysis and the provision of formal guarantees would have added significant value to our method. However, we believe that our experiments still demonstrate the heuristic validity of our approach, particularly in terms of robustness.
>
> We would also like to ask the reviewer to clarify what they mean by the “many easier ways” to reduce the effect of regularization (which, in our case, we do not aim to reduce, but to regularize). We believe that complexity provides a very intuitive approach, as it simply scales entropy by a measure of policy determinism.
>
> Finally, we are aware that the current applicability of our approach is limited to discrete action spaces. This was pointed out by us as a limitation in the Discussion section as well. However, we would like to underline the fact that several popular RL methods are limited to either discrete- or continuous-action spaces. Moreover, given the current wide applicability and inherent interest of discrete action spaces (e.g., language modeling, protein/molecular structuring, etc.), and the promising potential for extending complexity to continuous domains (by adapting the current disequilibrium formula to another distance metric between the policy and a deterministic one), we believe our approach already addresses highly relevant and impactful scenarios.

---

### Official Review · Reviewer_NpWr · 2025-11-11

**Soundness:** 2
**Presentation:** 1
**Contribution:** 1
**Rating:** 0
**Confidence:** 5

**Summary:**

This paper proposes Complexity-Driven Policy Optimization (CDPO), which replaces the standard entropy bonus in PPO with a “complexity” bonus defined as the product of Shannon entropy and disequilibrium (the deviation from a uniform distribution). This regularizer encourages exploration when the policy is nearly deterministic and imposes structure when it approaches pure randomness, alleviating the tedious tuning of the entropy coefficient and the tendency to collapse into uniform randomness. The authors conduct experiments on discrete-action tasks and show that CDPO is more robust to the regularization coefficient.

**Strengths:**

Originality: The contribution is incremental; it simply borrows the off-the-shelf LMC complexity measure without a deep analysis of why this particular form is superior in the context of the objective function.
Quality: The learning curves in the figures are presented in a rudimentary way and could be improved.
Clarity: Concepts, equations, pseudocode, figures, and detailed appendices are all provided, so reproducing the work is straightforward.

**Weaknesses:**

1. Disequilibrium relies on discrete action probabilities; the paper offers no clue on how to extend the method to continuous domains.
2. No convergence guarantees or regret bounds are given; the approach remains heuristic without theoretical evidence that complexity regularization is better than entropy.
3. Baselines are limited to PPO ± entropy; comparisons with other exploration-oriented methods such as SAC are missing, and all tested environments are rather simple.
4. It is unclear what benefits higher complexity brings. Although the paper analyzes the gradients of both entropy and complexity, the ultimate performance still depends on the environment’s objective function, leaving the relationship between complexity and algorithmic performance unexplained.

**Questions:**

See Weaknesses.

---

> ### Author Response · Authors · 2025-11-21
>
> We thank the reviewer for their comments and suggestions. However, we respectfully disagree in several ways with their comments. First of all, we thoroughly discussed the conceptual intuition behind the use of LMC complexity in place of entropy. We then provided a high-level analysis of their gradients to strengthen our intuition. While we acknowledge that no guarantee has been given to prove that complexity is superior, we provided experimental results confirming our claims. We would also like to understand why the learning curves have been considered as “rudimental”, and we hope to receive suggestions regarding how to improve them in the future.
>
> As far as the found weaknesses are concerned, please find our comments below.
>
> 1) We acknowledged this limitation in the paper, and we repeatedly stated that our work concerned discrete action spaces. We internally considered suggesting ways to extend this to continuous action spaces (e.g., by replacing disequilibrium with other distance metrics between the current policy and a deterministic one). However, we decided that, without experimental validation, it would be better not to make any conjectures. We will clarify this point more explicitly in future versions.
>
> 2) We acknowledge that the absence of convergence guarantees or regret bounds is a limitation, making our approach inherently heuristic. However, to the best of our knowledge, no convergence guarantees exist for PPO + entropy loss either, and the original papers advocating the use of entropy regularization do so based on empirical results, without theoretical guarantees [1–3].
>
> 3) As our goal was not to propose an exploration method but a regularization one, we chose not to compare it with exploration strategies, which remain orthogonal to ours and completely compatible with the use of complexity. Moreover, SAC was originally and primarily designed for continuous action spaces, and could, in theory, incorporate a complexity term instead of an entropy term. We decided to limit our analysis to the main discrete-action space algorithm and evaluate the impact of complexity vs entropy in that scenario. As for the environments, although some are relatively simple, we believe they encompass a variety of scenarios with different action spaces and require varying levels of entropy regularization, thus providing a meaningful testbed for our experiments. Testing our strategy on environments that are difficult for both PPO with entropy and PPO without entropy was out of the scope of our paper.
>
> 4) This is a general, yet straightforward, observation: the ultimate performance of any algorithm depends on the environment’s objective function. While entropy regularization can sometimes accelerate learning and other times slow it down (requiring careful tuning of its scaling factor) we have shown that complexity maximization has a consistently positive (or at least non-negative) effect on learning, regardless of its scaling factor. Its robustness to hyperparameters appears to us as a clear advantage of complexity.
>
> References:
>
> [1] Mnih, V., Badia, A.P., Mirza, M., Graves, A., Lillicrap, T., Harley, T., Silver, D. &amp; Kavukcuoglu, K. (2016). Asynchronous Methods for Deep Reinforcement Learning. *Proceedings of The 33rd International Conference on Machine Learning (ICML’16)*.
>
> [2] Schulman, J., Wolski, F., Dhariwal, P., Radford, A., & Klimov, O. (2017). Proximal policy optimization algorithms. *arXiv preprint arXiv:1707.06347*.
>
> [3] Williams, R.J. (1992). Simple statistical gradient-following algorithms for connectionist reinforcement learning. *Machine Learning* 8, 229–256.

---

### Meta-Review · Area_Chair_ypiN · 2026-01-02

**Summary:**

The reviewers raised the following concerns:

- Limited novelty: The contribution is incremental; it simply borrows the off-the-shelf LMC complexity measure without a deep analysis of why this particular form is superior in the context of the objective function.

- Lack theoretical justification: No theoretical evidences are provided to demonstrate that complexity regularization is better than entropy. No convergence guarantees or regret bounds are given.

- Limited empirical justification: Baselines are limited to PPO ± entropy; comparisons with other exploration-oriented methods and reward-based intrinsic motivation methods are missing, and all tested environments are rather simple. the difference between performance is very small.

- Lack deep analysis: It is unclear what benefits higher complexity brings. Although the paper analyzes the gradients of both entropy and complexity, the ultimate performance still depends on the environment’s objective function, leaving the relationship between complexity and algorithmic performance unexplained.

- Limited applications: The Applicability is limited to discrete action spaces; there is no clue on how to extend the method to continuous domains.

- Missing related work on other exploration-oriented methods.

- Missing details on the proposed method in Section 4.2.

- Negative effects of the proposed regularizarion: the complexity regularization might cause the policy to be neither structured nor explored enough.

**Reviewer Concerns:**

The following reviewer concerns have been partially addressed by the rebuttal:
- Limited novelty: although the substitution of entropy with complexity appear as a small modification, its conceptual impact is substantial, as it adapts entropy regularization based on the determinism of the current policy, representing a meaningful and original contribution.
- Missing comparison with other exploration-oriented methods: the main goal is not to propose an exploration strategy, but rather a regularization strategy that is orthogonal to and complementary with exploration strategies.
- Simple test environment: the goal was to compare its performance in terms of robustness, we had to stick with environments that we know could be solved by PPO with or without entropy. This led to the selection of simpler environments.
- Some unclear details and misunderstanding on the proposed method in Section 4.2.

The following concerns are still outstanding:
- Lack theoretical justification
- Limited empirical justification
- Lack deep analysis
- Limited applications
- Negative effects of the proposed regularization

**Reviewer Scores:**

Reviewer NpWr would keep the score because among his four concerns, the authors acknowledged two of them, and did not fully addressed the other two regarding limited justification and experiments.

Reviewer bMDS shares similar concerns as Reviewer NpWr, and would keep the score for the same reason.

Reviewer fukB might increase the score to 3 because the rebuttal addressed the concern on the unclear details on the proposed method.

Reviewer M85N would keep the score because most of his concerns still stand.

---

### Decision · Program_Chairs · 2026-01-26

Reject